# The Influence of Adherence to a Mediterranean Diet on Decompensation in Patients with Chronic Heart Failure

**DOI:** 10.3390/nu16193278

**Published:** 2024-09-27

**Authors:** Jose Jiménez-Torres, Carlos Jiménez-Juan, Ana Villa-Martínez, Marta Gutiérrez-García, Lourdes Moreno-Gaviño, Reyes Aparicio-Santos, Ángela Romero-Muñoz, M. José Goncet-Cansino, Bosco Barón-Franco, Máximo Bernabeu-Wittel

**Affiliations:** Department of Internal Medicine, University Hospital Virgen del Rocío, 41013 Seville, Spain; carlosjimjua@gmail.com (C.J.-J.); mbernabeu@us.es (M.B.-W.)

**Keywords:** heart failure, Mediterranean diet, adherence

## Abstract

Background: Chronic heart failure (CHF) is a major health problem, representing the main cause of hospitalization in people over 65 years of age. Several studies have associated the Mediterranean diet with a cardioprotective function, improving prognoses in patients with high cardiovascular risk. Our main objective is to determine whether higher adherence to the Mediterranean diet is associated with a lower severity of CHF, based on the number of decompensations and disease complications. Methods: This study was a single-center retrospective cohort study conducted at the Virgen del Rocío Hospital (Seville). Adherence to a Mediterranean diet was determined by the Mediterranean Diet Adherence Screener (MEDAS) in patients with chronic heart failure in a state of clinical stability, the number of decompensations in the 12 months before inclusion, cardiac biomarkers (NT-proBNP and CA125), evaluation of dyspnea, and quality of life assessment according to NYHA and KCCQ scales and analytical profiles. Results: Seventy-two patients were included (35 with high adherence to the Mediterranean diet and 37 with low adherence). The mean age was 81.29 ± 0.86 years. A trend towards fewer decompensations (1.49 ± 0.14 vs. 1.92 ± 0.17, *p* = 0.054) and lower NT-proBNP values (2897.02 ± 617.16 vs. 5227.96 ± 1047.12; *p* = 0.088) was observed in patients with high adherence compared to those with low adherence to the Mediterranean diet. Conclusions: Our results suggest that patients with CHF and high adherence to the Mediterranean diet have a tendency towards an improved cardiac profile, indicated by fewer decompensations and lower NT-proBNP levels. Future clinical trials are needed to substantiate these hypotheses.

## 1. Introduction

Chronic heart failure (CHF) is the leading cause of hospitalization in patients over 65 years old and presents a high readmission rate within 30 days post-discharge [1]. The prevalence of HF is estimated at 1–2% of the adult population in developed countries, increasing proportionally with age, reaching over 10% in patients older than 70 years [2].

Diet plays a fundamental role in lifestyle, but recommendations for HF are not well defined. Most evidence is based on sodium restriction in these patients, which is difficult to interpret due to the variability in study designs. Lara K. et al. [3] demonstrated that a diet based on plant products is associated with a lower risk of HF and HF-related hospitalizations, contrary to those with a high intake of red and processed meats, sugary drinks, and refined flours, who have a higher risk of HF. The Mediterranean diet has been associated with a reduction in cardiovascular events, as shown in the CORDIOPREV study [4], which identified a 33% reduction in the incidence of cardiovascular events compared to a low-fat diet in patients with coronary disease after 7 years of follow-up.

The primary objective of this study was to determine if higher adherence to a Mediterranean diet pattern is associated with a lowered severity of HF, indicated by reduced cardiac decompensations in the previous 12 months, a better functional class according to the New York Heart Association (NYHA) scale, higher perceived quality of life according to the Spanish version of the Kansas City Cardiomyopathy Questionnaire (KCCQ), and lower plasma concentrations of congestive biomarkers (NTproBNP and CA125) compared to low adherence to a Mediterranean diet.

## 2. Methods

The present study was a single-center retrospective cohort study conducted in the comprehensive Internal Medicine Unit at Virgen del Rocío Hospital (Seville).

Patients with a previous diagnosis of CHF, clinically stable for 1 month since the last episode of decompensations, were included. Patients with other advanced or uncontrolled chronic diseases or those with a Barthel index < 60 points were excluded.

The primary variables included the number of decompensations in the 12 months before study inclusion (including hospitalizations, emergency care, or intravenous diuretic use in specialized clinics), the degree of dyspnea assessed through the NYHA scale, the 12-item short form of the Kansas City Cardiomyopathy Questionnaire (KCCQ), and the determination of serum biomarkers (NTproBNP and CA125). Secondary variables included anthropometric measurements, gender, age, treatments for managing cardiovascular risk factors, and previous comorbidities. Adherence to a Mediterranean diet was determined using the MEDAS questionnaire [5]. A score of ≥9 points corresponded to high adherence to the Mediterranean diet.

Statistical analysis was conducted using SPSS (version 23.0 for Windows) (SPSS Inc., Chicago, IL, USA). Descriptive statistics were detailed as numbers (and percentages (%)) for qualitative variables and as mean ± standard error of the mean (SE) for quantitative variables, depending on the distribution. The distribution of quantitative variables was evaluated using the Kolmogorov–Smirnov test. To detect differences between groups, Chi-square tests (Fisher’s test when necessary) and Student’s *t*-test (Mann–Whitney U test in case of non-normal distribution) were used. To determinate the contribution of high adherence to a Mediterranean diet to the reduction in decompensations, we performed a multiple linear regression using the number of decompensations as the dependent variable. Ever smoking, high adherence to the Mediterranean diet, and left ventricular ejection fraction were included in the analysis, assuming that all predictor variables were quantitative or categorical (with two categories), and the outcome variable was quantitative, continuous, and unbounded. 

Differences were quantified using the odds ratio and the difference in means (or ranks) with 95% confidence intervals. The level of statistical significance was set at *p* < 0.05 for two tails. The project was approved by the Clinical Research and Ethics Committee of the Virgen del Rocío—Virgen Macarena University Hospital.

## 3. Results

In total, 201 patients were evaluated consecutively, of whom 129 were excluded for not meeting the inclusion criteria or refusing to participate. A final total of 72 patients were included, of which 37 had low adherence to a Mediterranean diet and 35 had high adherence. The average age was 81.29 ± 0.86 years, and 59.7% were women. There was a higher number of patients with chronic kidney disease and type 2 diabetes, as well as lower LDL cholesterol concentrations, in patients with low adherence (*p* < 0.05). The rest of the baseline characteristics and the comparison between both groups are shown in Table 1. 

Regarding treatments, patients with high adherence had lower use of lipid-lowering drugs; the rest of the therapeutic groups can be seen in Table 2. The number of decompensations was 1.92 ± 0.17 in the low adherence group vs. 1.49 ± 0.14 (*p* = 0.054) in the high adherence group, while HF hospitalization was 1.27 ± 0.17 in the low adherence group vs. 1.00 ± 0.10 in the high adherence group (*p* = 0.188).

No differences were observed in KCCQ scores (67.35 ± 3.32 in the low adherence group vs. 69.92 ± 3.24 in the high adherence group, *p* = 0.524) or NYHA scores (*p* = 0.207).

The average levels of NTproBNP were 4094.87 ± 627.28 pg/mL (5227.96 ± 1047.12 in patients with low adherence vs. 2897.02 ± 617.16 in patients with high adherence, *p* = 0.088), while CA125 values were 43.06 ± 8.81 U/mL (53.30 ± 16.32 and 33.28 ± 5.44, *p* = 0.973, respectively).

In a stepwise multiple linear regression analysis using the number of decompensations as the dependent variable, ever smoking, high adherence to a Mediterranean diet (vs. low adherence), and left ventricular ejection fraction were significant contributors (*p* < 0.05) to reducing decompensations in patients with CHF (Table 3). 

## 4. Discussion

The results of this study suggest that HF patients following a Mediterranean diet tend to have a better cardiac profile, indicated by fewer decompensations and lower NTproBNP levels, without statistically significant differences compared to HF patients with low adherence to a Mediterranean diet. Additionally, our study shows that these patients have a lower risk of type 2 diabetes or chronic kidney disease, with lower use of lipid-lowering drugs, although they had higher LDL cholesterol levels.

The MEDIT-AHF study [6], an observational study that included 991 patients with a previous diagnosis of acute heart failure, reported that the number of HF decompensations was not significantly related to the Mediterranean diet (*p* = 0.49) after 1 year of follow-up. However, the hospitalization rate for HF was lower in the Mediterranean diet adherence group compared to the non-adherent group, with a 26% risk reduction. The differences could be explained by the larger sample size in the MEDIT-AHF study compared to the present study.

The benefits of the Mediterranean diet on the body could influence the reduction in decompensations suggested in this study. This has justified the reduction in the number of hospitalizations in other studies, although the underlying mechanism involved in this hypothesis is not defined. The Mediterranean diet has demonstrated cardiovascular benefits from the consumption of fruits, vegetables, and monounsaturated fats from extra virgin olive oil and nuts, which help reduce insulin resistance, improve serum glucose, increase HDL cholesterol levels, reduce blood pressure, and decrease oxidative stress. Also, it has been observed that it can improve diastolic function on echocardiography and cardiorespiratory fitness measured by maximum oxygen consumption, which could improve cardiac contractility [3,7,8,9,10].

Furthermore, a sub-analysis of the PREDIMED study [10], which included 930 patients with high cardiovascular risk, observed a decrease in inflammatory markers and prognostic biomarkers in the development of HF (such as NT-proBNP) in patients adhering to a Mediterranean diet. This study showed a significant reduction in this marker associated with Mediterranean diet adherence in the group supplemented with extra virgin olive oil (*p* = 0.029) and the group consuming nuts (*p* = 0.006). These results translate to a lower risk of hospitalization in HF patients, as NT-proBNP has proven to be very useful in assessing the risk of readmission and short-term mortality. Studies have shown that the variability of these values indicates the severity and prognosis of HF after treatment, so their decrease is associated with a lower risk of hospitalization [11,12]. Therefore, the findings of this study suggest some benefit on NT-proBNP levels and the risk of congestive HF. 

Patients with high adherence to a Mediterranean diet had higher LDL cholesterol levels, as well as a lower consumption of lipid-lowering drugs, compared to low-adherence patients. The study did not evaluate the target LDL level of each patient, allergies, or the intensity of statin therapy, so we cannot conclude whether these differences are due to inadequate lipid-lowering treatment or to the influence of the diet, among other reasons.

The limitations of this study include the retrospective design and a small population size, so clinical trials are required to establish the relationship between the Mediterranean diet and HF.

## 5. Conclusions

Our results suggest that high adherence to the Mediterranean diet in patients with CHF tends to improve the cardiac profile, indicated by a reduced number of decompensations and lower NT-proBNP levels, without differences in hospitalization needs for HF, degree of dyspnea, or functional capacity. Future clinical trials are needed to substantiate these hypotheses.

## Figures and Tables

**Table 1 nutrients-16-03278-t001:** Study population characteristics based on a low or high adherence to a Mediterranean diet.

	All Patients *n* = 72	Low Adherence *n* = 37	High Adherence *n* = 35	*p* Value
Age (years)	81.29 ± 0.86	80.95 ± 1.11	81.66 ± 1.33	0.456
Gender: Female, *n* (%)	43 (59.7)	18 (48.6)	25 (71.4)	0.049 *
T2DM, n (%)	39 (54.2)	25 (67.6)	14 (40)	0.019 *
Coronary heart disease, *n* (%)	21 (29.2)	13 (35.1)	8 (22.9)	0.252
peripheral arterial disease, *n* (%)	6 (8.3)	4 (10.8)	2 (5.7)	0.434
COPD, *n* (%)	16 (22.2)	6 (16.2)	10 (28.6)	0.208
Liver disease, *n* (%)	6 (8.3)	1 (2.7)	5 (14.3)	0.076
CKD, *n* (%)	41 (56.9)	26 (70.3)	15 (42.9)	0.019 *
Barthel	89.1 ± 1.39	89.32 ± 2.08	88.86 ± 1.86	0.515
Systolic blood pressure (mmHg)	127.16 ± 2.2	126 ± 3.31	128.4 ± 2.9	0.589
Diastolic blood pressure (mmHg)	65.65 ± 1.7	65.59 ± 2.32	65.71 ± 2.52	0.972
BMI	28.48 ± 0.63	27.77 ± 0.69	29.24 ± 1.08	0.256
Abdominal circumference (cm)	103.2 ± 1.31	103.39 ± 1.51	103 ± 2.19	0.731
MEDAS	8.51 ± 0.26	6.81 ± 0.22	10.31 ± 0.24	<0.001 *
Reduced LVEF, *n* (%)	19 (26.4)	11 (29.7)	8 (22.9)	0.508
atrial fibrillation, *n* (%)	52 (72.2)	27 (75)	25 (73.5)	0.888
Time to decompensation–inclusion (days)	116 ± 13.31	105.81 ± 18.64	126.69 ± 19.14	0.219
Hemoglobin (g/dL)	12.93 ± 0.23	12.88 ± 0.33	12.99 ± 0.33	0.816
LDL (mg/dL)	84.9 ± 4.75	71.89 ± 6.79	98.65 ± 5.88	0.004 *
HDL (mg/dL)	43.02 ± 2.02	38.78 ± 3.18	47.51 ± 2.24	0.112
Triglycerides (mg/dL)	112.23 ± 7.01	110.35 ± 12	114.22 ± 7.05	0.782
Ferritin (ng/mL)	199.18 ± 25.51	178.47 ± 29.26	221.08 ± 42.54	0.640
Creatinine (mg/dL)	1.37 ± 0.07	1.50 ± 0.12	1.26 ± 0.08	0.239
Glomerular Filtrate (mL/min/1.73 m^2^)	48.35 ± 2.6	46.39 ± 4.02	50.9 ± 3.60	0.352
hsCRP (mg/mL)	11.95 ± 2.17	15.75 ± 3.81	7.80 ± 1.61	0.732

The number of patients with each characteristic is shown together with the percentage (%) or the mean values ± standard error. We used unpaired *t* tests for quantitative variables and χ^2^ for categorical variables. T2DM: type 2 diabetes mellitus; COPD: chronic obstructive pulmonary disease; CKD: chronic kidney disease; BMI: body mass index; MEDAS: Mediterranean Diet Adherence Screener; LVEF: left ventricular ejection fraction; LDL: low-density lipoprotein; HDL: high-density lipoprotein; hsCRP: ultra-sensitive C-reactive protein. * *p* < 0.05 HF patients with high adherence to a Mediterranean diet vs. HF patients with low adherence to a Mediterranean diet.

**Table 2 nutrients-16-03278-t002:** Baseline medication based on a low or high adherence to a Mediterranean diet.

	All Patients *n* = 72	Low Adherence *n* = 37	High Adherence *n* = 35	*p* Value
Antihypertensive, *n* (%)	67 (93.1)	36 (97.3)	31 (88.6)	0.145
ACE inhibitor/ARB, *n* (%)	35 (48.6)	20 (54.1)	15 (42.9)	0.342
ARNI, *n* (%)	8 (11.1)	5 (13.5)	3 (8.6)	0.505
Calcium antagonist, *n* (%)	9 (12.5)	5 (13.5)	4 (11.4)	0.789
Beta-blocker, *n* (%)	62 (86.1)	31 (83.8)	31 (88.6)	0.557
Diuretic, *n* (%)	70 (97.2)	37 (100)	33 (94.3)	0.140
Loop diuretic, *n* (%)	64 (88.9)	34 (91.9)	30 (85.7)	0.404
SGLT2 inhibitor, *n* (%)	53 (73.6)	25 (67.6)	28 (80)	0.232
MRA, *n* (%)	34 (47.2)	19 (51.4)	15 (42.9)	0.471
Thiazide, *n* (%)	13 (18.1)	8 (21.6)	5 (14.3)	0.419
Acetazolamide, *n* (%)	1 (1.4)	1 (2.8)	0 (0)	0.321
Lipid-lowering drugs, *n* (%)	42 (58.3)	26 (70.3)	16 (45.7)	0.035 *
Statin, *n* (%)	41 (56.9)	25 (67.6)	16 (45.7)	0.061
Fibrate, *n* (%)	1 (1.4)	1 (2.8)	0 (0)	0.327
Other	10 (13.9)	4 (10.8)	6 (17.1)	0.437
Antidiabetic, *n* (%)	20 (28.2)	11 (29.7)	9 (26.5)	0.760
Metformin, *n* (%)	11 (15.5)	6 (16.2)	5 (14.7)	0.861
GLP-1RA, *n* (%)	9 (12.7)	4 (10.8)	5 (14.7)	0.622
Insulin, *n* (%)	10 (14.1)	5 (13.5)	5 (14.7)	0.885

The number of patients taking each drug is shown alongside the percentage (%). We used unpaired *t* tests for quantitative variables and χ_2_ for categorical variables. ACE inhibitor: Angiotensin Converting Enzyme Inhibitor; ARB: Angiotensin-receptor blocker; ARNI: Angiotensin-Neprisylin Receptor inhibitor; SGLT2: sodium–glucose cotransporter type 2; GLP-1RA: Glucagon-like peptide-1 receptor agonist; MRA: mineralocorticoid receptor antagonist. * *p* < 0.05 HF patients with high adherence to a Mediterranean diet vs. HF patients with low adherence to a Mediterranean diet.

**Table 3 nutrients-16-03278-t003:** Statistically significant multiple linear regression coefficients to predict heart failure decompensation.

Independent Variables	Unstandardized Coefficients		Standardized Coefficients	*p* Value
	B	SE		
Left ventricular ejection fraction, %	0.022	0.07	0.349	0.002
High adherence to a *Mediterranean diet (vs. low adherence)* ^†^	−0.481	0.213	−0.251	0.027
Ever smoking, *yes*	0.259	0.115	0.252	0.027

Predictive variables tested by stepwise method: age, gender, ever smoking, body mass index, presence of type 2 diabetes mellitus, high adherence (vs. low adherence) to a Mediterranean diet, pharmacological treatments (use of lipid-lowering drugs, use of antidiabetic drugs, and use of antihypertensive drugs, use of diuretic drugs), left ventricular ejection fraction, body mass index, presence of chronic kidney disease, chronic obstructive pulmonary disease, peripheral arterial disease, hemoglobin, ferritin, coronary disease, low-density lipoprotein cholesterol and glomerular filtrate, assuming that all predictor variables were quantitative or categorical (with two categories), and the outcome variable was quantitative, continuous and unbounded. High adherence to a *Mediterranean diet*
^†^ contributed to a reduction in congestive heart failure in this model. B: Beta; SE: standard error.

## Data Availability

The raw data supporting the conclusions of this article will be made available by the authors on request.

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
