# Peer review of "The Influence of Adherence to a Mediterranean Diet on Decompensation in Patients with Chronic Heart Failure"

_nutrients, 2024, doi:10.3390/nu16193278_

Round 1

Reviewer 1 Report

Comments and Suggestions for Authors

In the study by Jose Jiménez-Torres, they investigated “INFLUENCE OF ADHERENCE TO A MEDITERRANEAN DIET ON DESCOMPENSATION IN PATIENTS WITH CHRONIC HEART FAILURE”. Diet, as an external factor ingested orally, is often perceived as being related to health and disease.

However, blinded RCTs targeting foods are difficult to conduct, and observational studies are preferred. Compared to articles on substances such as tobacco and alcohol, the relationship between diet and health / disease is prone to bias, making RCTs theoretically challenging to perform. This study examined the relationship between the Mediterranean diet, which has long been associated with lifespan and health and has relatively well-established research methods, and heart failure.

Despite this, there are some issues that must be addressed.

Concern #1: There appears to be no significant difference in the primary variables investigated. This finding seems to differ from past studies mentioned in the Discussion section of this article. Have you considered this discrepancy?

Concern #2: When was the Mediterranean diet adherence survey conducted? Was it during hospitalization? At the time of study participation? Was the change in adherence rate monitored during the observation period?

Concern #3: The low adherence group had a higher rate of ischemic heart disease compared to the high adherence group, although this difference was not significant. The low adherence group had significantly higher rates of diabetes mellitus (DM), males, and chronic kidney disease (CKD), while the high adherence group had higher LDL levels. This difference is very interesting. Would the results be the same if you analyzed ischemic heart disease alone? This data is worth investigating and would improve the quality of the paper.

Concern #4: This study defines heart failure as stable for at least one month. The average seems to be about 100 days. Can you define stability at one month? Please investigate the time until re-admission for patients who were readmitted.

Concern #5: The results of the primary variables in this study are only stated in the text. Even though there is no significant difference, the paper is difficult to recognize this outcome. I recommend including a Kaplan-Meier curve with heart failure treatment intervention and heart failure hospitalization as outcomes.

Author Response

Thank you for the time you took to review this manuscript. Your comments are very helpful and contribute to the quality of the text. I hope that my responses and modifications will be to your liking and will improve the initial version.

Comments 1: There appears to be no significant difference in the primary variables investigated. This finding seems to differ from past studies mentioned in the Discussion section of this article. Have you considered this discrepancy?

Response 1: Thank you very much for your comment. In fact, there is a certain difference with respect to previous studies. Probably, the small sample size compared to the other studies is the cause of these small differences. The same tendency remains in spite of this. In order to emphasize this better, the discussion has been modified to highlight these differences.

Comments 2: When was the Mediterranean diet adherence survey conducted? Was it during hospitalization? At the time of study participation? Was the change in adherence rate monitored during the observation period?

Response 2: The evaluation of the patient and the survey of adherence to the Mediterranean diet were carried out at a single time, during visits. It was required that they were in a stable phase.

For the analysis of cardiac events, the 12 months previous to the date of inclusion were evaluated.  

Comments 3. The low adherence group had a higher rate of ischemic heart disease compared to the high adherence group, although this difference was not significant. The low adherence group had significantly higher rates of diabetes mellitus (DM), males, and chronic kidney disease (CKD), while the high adherence group had higher LDL levels. This difference is very interesting. Would the results be the same if you analyzed ischemic heart disease alone? This data is worth investigating and would improve the quality of the paper.

Response 3. Thank you for your comment. The study is limited by the sample size. So in order to assess that other conditions do not influence the differences observed in the study, a multivariate analysis has been performed. Ischemic heart disease does not significantly influence the fitted model. This change can be found the revised manuscript in page 8, table 3,

Comment 4.  This study defines heart failure as stable for at least one month. The average seems to be about 100 days. Can you define stability at one month? Please investigate the time until re-admission for patients who were readmitted.

Response 4. The study has a retrospective design, so readmissions after the time of inclusion in the study cannot be defined.

Cardiac stability was defined as stability in the usual level of dyspnea and absence of worsening edema or need for higher doses of diuretics. At least 30 days were required in this situation. The presence of patients in both groups with long periods of stability probably influenced the time observed.

Comment 5. The results of the primary variables in this study are only stated in the text. Even though there is no significant difference, the paper is difficult to recognize this outcome. I recommend including a Kaplan-Meier curve with heart failure treatment intervention and heart failure hospitalization as outcomes.

Response 5. Thank you for your appreciation and suggestion. It may be that performing kaplan-meier curve may improve the visualization of the observed results. Unfortunately, due to lack of time and away to be able to review the dates of decompensations, it is not possible to perform this analysis.

Reviewer 2 Report

Comments and Suggestions for Authors

1. Did they perform any adjustments or multiple regression analyses on the primary results? How can we be sure that differences between the two groups do not influence the results?

2. In the Abstract section:

- “average” is not the most appropriate term in medical statistics;

- to make this section more relevant, supplementary details on the methods are required: the duration of the retrospective analysis, the use of the NYHA scale, congestion biomarkers, and KCCQ-derived data.

3. Abbreviations in the two tables and their legends should use standardised, English-derived abbreviations. Acronyms such as DM2, ARAII, iSGLT2, and ARM should be replaced by the equivalent, usual English abbreviations. “Arteriopatia” is not an English term either.

4. What does the GLP1 abbreviation refer to? Is it referring to GLP-1 receptor agonists (the correct name of this drug category, which should be abbreviated as GLP-1 RA) or to the larger class of incretin-based drugs, dipeptidyl peptidase-4 inhibitors included?

5. The authors should comment on the lower proportion of statin users among patients adherent to the Mediterranean diet. Is it because this dietary pattern improves the lipid profile? On the other hand, statins exert extended pleiotropic benefits, making them necessary for secondary prevention, irrespective of the serum lipids level, or even primary prevention of patients with elevated atherosclerotic cardiovascular risk.

6. In the Discussion section: this study has a retrospective design, not a cross-sectional design.

7. Please improve the References style (eliminate the double numbering and add the missing page ranges and/or DOIs).

Comments on the Quality of English Language

The whole manuscript would benefit from a thorough English language assessment. A series of typing and syntactic errors (e.g., inappropriate use of capital letters and prepositions, missing spaces or punctuation marks, inaccurate orthographies, and some unnatural English constructs) undermine the background of an otherwise acceptable quality of the English language.

Author Response

Thank you for taking the time to review this manuscript. Your comments I agree with and are the result of a certain immaturity in article writing. You will find detailed responses and corrections below. The quality of the English has been checked as best as possible and the revised text has been modified accordingly. 
I hope you will find it to your satisfaction.

Comments 1: Did they perform any adjustments or multiple regression analyses on the primary results? How can we be sure that differences between the two groups do not influence the results?

Response 1: Thank you for pointing this out. We have carried out a “step by step” multivariate analysis. Although "R" is low, high adherence Mediterranean diet contributed to a reduction of number of descompensations in this model. The revised manuscript this change can be found on page number 8m “table 2”.

Comments 2: In the Abstract section:- “average” is not the most appropriate term in medical statistics;- to make this section more relevant, supplementary details on the methods are required: the duration of the retrospective analysis, the use of the NYHA scale, congestion biomarkers, and KCCQ-derived data.

Response 2: Agree. Consequently, I have modified the summary in the "methods" section to make it more visible and the term has been modified according to.  This change can be found the revised manuscript on page 1, paragraph “methods”.

Comments 3. Abbreviations in the two tables and their legends should use standardised, English-derived abbreviations. Acronyms such as DM2, ARAII, iSGLT2, and ARM should be replaced by the equivalent, usual English abbreviations. “Arteriopatia” is not an English term either.

Response 3. I regret the error in the transcription of the table. The corresponding abbreviations have been modified to make it understandable in English abbreviations. This change can be found the revised manuscript on page 6 (table 1, legend), page 7 (table 2, legend)

Comment 4.  What does the GLP1 abbreviation refer to? Is it referring to GLP-1 receptor agonists (the correct name of this drug category, which should be abbreviated as GLP-1 RA) or to the larger class of incretin-based drugs, dipeptidyl peptidase-4 inhibitors included?

Response 4. Thank you for this comment. GLP-1 referring to GLP-1 RA. This change can be found the revised manuscript on page 6 (table 2).

Comment 5. The authors should comment on the lower proportion of statin users among patients adherent to the Mediterranean diet. Is it because this dietary pattern improves the lipid profile? On the other hand, statins exert extended pleiotropic benefits, making them necessary for secondary prevention, irrespective of the serum lipids level, or even primary prevention of patients with elevated atherosclerotic cardiovascular risk.

Response 5. It is true that, based on the observational nature of the study, it was not controlled whether the patients were on therapeutic targets, or had allergies or contraindications. There is an inertia to discontinue treatment in elderly patients, which may also have influenced this finding. In accordance with your comment, the text in the "discussion" section has been modified: "...The high adherence patients had higher LDL cholesterol levels, as well as a lower consumption of lipid-lowering drugs compared to low adherence patients to the Mediterranean diet. The study did not evaluated the target LDL level of each patient, allergies or the intensity of statin therapy, so we cannot conclude whether these differences are due to inadequate lipid-lowering treatment or to the influence of diet, among other reasons. ....".

Comment 6. In the Discussion section: this study has a retrospective design, not a cross-sectional design.

Response 6. Agree. Consequently, I have modified the text. This change can be found the revised manuscript on page 4, line 12.

Comment 7. Please improve the References style (eliminate the double numbering and add the missing page ranges and/or DOIs).

Response 7. Thank you very much for your appreciation. I have reviewed the bibliography and it has been adapted according to the required standards. You can see this change in section "references".

Reviewer 3 Report

Comments and Suggestions for Authors

This is a well and professionally written paper.

The topic is of clinical interest and warrants attention, in particularly in this elderly patient (81(SD1.0)years)group.

I do not understand the math of the first sentence of the Results section on page 2 (Line 78, 81). It reads << Of the 371 patients consecutively evaluated in medical consultations, 129 were excluded for not meeting the inclusion criteria or declining participation. Seventy-two patients were included, of which 37 had low adherence to a Mediterranean diet and 35 had 80 high adherence >>.

To me 371 evaluated minus 129 excluded makes 142 eligible subjects.

Thus, I do not understand why only 72 of the 142 eligible patients were included in the study and statistical analyses. Were these 72 subjects the (only) ones that adhered to the (high and low adherence) Med diet? Were of the142 patients (for example those with a Barthel index < 60 points)  excluded in the observation process not included in the analysis, leaving 72 patients? This is unclear.

The major shortcoming is the remaining sample size (72, or 35 on high and 35 subjects on high adherence diet) for the primary endpoint (HF) showing trends, where if the population would have been double the size (still small) suggestion of efficacy would turn into a fact.

Remarks:

The paper, except for the first lines of the results, is well written and reads well.

This is a very important and relevant study topic and the results indicate (but not prove) the sense of high adherence to Med diet even at high age.

Suggestions:

As is the sample is too small. The sample must be extended to draw solid conclusions. The sample may include the present data.

It is of clinical relevance to include matched group not on Med diet.

The lower statin use in the high Med diet group may need some extra attention and considerations in the a Discussion section of the next version of the manuscript. Have the authors any clue what is causing this?

I hope the authors will extent the sample and study design to address these important topics.

Author Response

Dear reviewer. I thank you for the time dedicated to reviewing this manuscript, as well as your contributions. We have taken into account all your comments and we agree with them. You can see the responses to your comments below.

Comments 1: As is the sample is too small. The sample must be extended to draw solid conclusions. The sample may include the present data. 

It is of clinical relevance to include matched group not on Med diet.

Response 1: Thank you for your comment. Indeed, we think that increasing the sample size would improve the results obtained and would be in line with previous studies, despite the advanced age of this study.

The lack of time to increase the population size is the reason for not being able to send you other data.

Regarding following another diet, it would require greater control by researchers and the use of other questionnaires, such as food frequency questionnaires. This could be better controlled in dietary intervention studies.

Comments 2: I do not understand the math of the first sentence of the Results section on page 2 (Line 78, 81). It reads << Of the 371 patients consecutively evaluated in medical consultations, 129 were excluded for not meeting the inclusion criteria or declining participation. Seventy-two patients were included, of which 37 had low adherence to a Mediterranean diet and 35 had 80 high adherence >>.

To me 371 evaluated minus 129 excluded makes 142 eligible subjects.

Thus, I do not understand why only 72 of the 142 eligible patients were included in the study and statistical analyses. Were these 72 subjects the (only) ones that adhered to the (high and low adherence) Med diet? Were of the142 patients (for example those with a Barthel index < 60 points)  excluded in the observation process not included in the analysis, leaving 72 patients? This is unclear.

Response 2: Thank you very much for your comment. In effect, this is an error when transcribing the flow chart.

371 patients were scheduled for consultations, but 201 patients were eligible for the study (face-to-face review, possibility of accessing data prior to their assessment), of which 129 patients were excluded.Consequently, the paper has been modified, and the correction can be seen on page 2, line 2.

Comments 3. The lower statin use in the high Med diet group may need some extra attention and considerations in the a Discussion section of the next version of the manuscript. Have the authors any clue what is causing this?

Response 3.  It is true that, based on the observational nature of the study, it was not controlled whether the patients were on therapeutic targets, or had allergies or contraindications. There is an inertia to discontinue treatment in elderly patients, which may also have influenced this finding. In accordance with your comment, the text in the "discussion" section has been modified: "...The high adherence patients had higher LDL cholesterol levels, as well as a lower consumption of lipid-lowering drugs compared to low adherence patients to the Mediterranean diet. The study did not evaluated the target LDL level of each patient, allergies or the intensity of statin therapy, so we cannot conclude whether these differences are due to inadequate lipid-lowering treatment or to the influence of diet, among other reasons. ....".

Round 2

Reviewer 1 Report

Comments and Suggestions for Authors

The authors have addressed essentially all my previous comments, and their revisions have substantially improved the manuscript. I have no further comments.

Author Response

Dear reviewer,

We thank you for your comments and suggestions for improving the manuscript. The authors have considered the comments and revised the manuscript accordingly.

Grammar and some terms have been modified in the text, according to other reviewer considerations.

Reviewer 2 Report

Comments and Suggestions for Authors

Compared to the previous version of the manuscript, the current form of the paper shows some improvements. However, some issues have not been sufficiently approached:

1. The authors claim they have “carried out a step-by-step multivariate analysis”. Yet, the Methods section mentions nothing about this procedure, and the final section of the Results (lines 122-132 and Table 3) includes too few details to clarify this aspect.

2. Abbreviations in Table 3 are not explained in the corresponding legend.

3. The manuscript still requires correction of:

- Erroneous use of medical terms, most of which are probably due to a tendency to translate idioms from their native language literally. Just a few examples: “descompensation”, “peripheral arteriopathy disease”, “glucagon-like peptide type 1 receptor agonists”. Unfortunately, such mistakes are still frequent throughout the manuscript and do not seem to have been appropriately approached by the authors.

Comments on the Quality of English Language

- Part of the syntactic errors I already warned the authors about (e.g., inaccurate topics or constructs, wrong verbal forms, inappropriate use of capital letters and prepositions) were not yet corrected. A few examples here: “lower cardiac decompensations”, “hospitalizations for it”, “before to study inclusion”, “Two hundred one patients consecutively evaluated in medical consultations, of which 129 were excluded for not meeting the inclusion criteria or declining participation”, “smoke” instead of “smoking”, etc.

Author Response

Dear reviewer,

We thank you for your comments and suggestions for improving the manuscript. The authors have considered the comments and revised the manuscript accordingly. Thank you very much. We replied your comments as below:

Comment 1. The authors claim they have “carried out a step-by-step multivariate analysis”. Yet, the Methods section mentions nothing about this procedure, and the final section of the Results (lines 122-132 and Table 3) includes too few details to clarify this aspect.

Response. Thank you for this pertinent comment. We agree with the reviewer. We have modificated Methods and Results sections to clarify this. You can see these on page 2, lines 43-48 ("...To determinate the contribution of high adherence Mediterranean diet to reduction of congestive heart failure, we performed a multiple linear regression using the number of congestive heart failure as the dependent variable. Ever smoking, high adherence Mediterranean diet and left ventricular ejection fraction were included in the analysis, assuming that all predictor variables were quantitative or categorical (with two categories), and the outcome variable was quantitative, continuous and unbounded." and page 3, lines 24-28 ("...in a stepwise multiple linear regression analysis using the number of congestive HF as the dependent variable, ever smoking, high adherence Mediterranean diet (vs low adherence) and left ventricular ejection fraction were significant contributors (p<0.05) to reduce congestive heart failure in patients with chronic heart failure (Table 3)...".

Comment 2.  Abbreviations in Table 3 are not explained in the corresponding legend.

Response. There are no abbreviations in the table legend after revision of the text.

Comment 3. 

The manuscript still requires correction of:

  • Erroneous use of medical terms, most of which are probably due to a tendency to translate idioms from their native language literally. Just a few examples: “descompensation”, “peripheral arteriopathy disease”, “glucagon-like peptide type 1 receptor agonists”. Unfortunately, such mistakes are still frequent throughout the manuscript and do not seem to have been appropriately approached by the authors.

Response. 

We appreciate this important observation pointed out by the reviewer. According to that, we have modified these terms. "Descompensation" has been modified to “congestive heart failure”, “peripheral arteriopathy disease” has been modified to "“peripheral arterial disease”, “glucagon-like peptide type 1 receptor agonists” has been modified to “glucagon-like peptide-1 receptor agonists”. You can see these on the revised manuscript. 

Comments on the Quality of English Language
  • Part of the syntactic errors I already warned the authors about (e.g., inaccurate topics or constructs, wrong verbal forms, inappropriate use of capital letters and prepositions) were not yet corrected. A few examples here: “lower cardiac decompensations”, “hospitalizations for it”, “before to study inclusion”, “Two hundred one patients consecutively evaluated in medical consultations, of which 129 were excluded for not meeting the inclusion criteria or declining participation”, “smoke” instead of “smoking”, etc.

We are totally agreeing with the reviewer. In this sense, the text has been revised and the errors detected have been modified.

Reviewer 3 Report

Comments and Suggestions for Authors

Thank you for allowing me to review the revised manuscript and the provided answers.

As stated previously, the manuscript is very well written and reads well. My concerns remain, however as the authors pragmatically state, the sample size, cannot be altered. The latter is not a scientific argument, but the topic is of importance and clinical relevance, and the quality of the presentation is high. We may view this retrospective work as a stepping stone for further research that is worth publishing.

As is, I have no further comments at this point. 

Author Response

Dear reviewer,

Thank you for your efforts and considerations for improving the manuscript. We agree that a controlled study may be the best way to ensure these results.
Grammar and some terms have been modified in the text, according to other considerations.

Round 3

Reviewer 2 Report

Comments and Suggestions for Authors

1. I did not advise the authors to avoid the term “decompensation” completely. It is not incorrect in English, and I have only objected to its frequent misspelling as “descompensation”, which was not corrected after the first revision. I even think that this term would often have shed a more precise light on the paper's focus than the term “congestive heart failure”, which does not reflect precisely enough the acute episodes of decompensation the authors evaluated.

2. I still think the B and SE abbreviations in Table 3 need an explanation in the associated legend.

Comments on the Quality of English Language

3. Most English language errors have been approached (see above).

Author Response

Dear reviewer, We thank you for your comments and suggestions for improving the manuscript. The authors have considered the comments and revised the manuscript accordingly. Thank you very much. We replied your comments as below:

Comment 1:  I did not advise the authors to avoid the term “decompensation” completely. It is not incorrect in English, and I have only objected to its frequent misspelling as “descompensation”, which was not corrected after the first revision. I even think that this term would often have shed a more precise light on the paper's focus than the term “congestive heart failure”, which does not reflect precisely enough the acute episodes of decompensation the authors evaluated.

Response: Dear reviewer, we are sorry we did not understand you. In our opinion, “decompensation” is the ideal term, and “congestive heart failure" is close but not ideal. The text has been revised again and the grammatical errors have been corrected.

Comment 2. I still think the B and SE abbreviations in Table 3 need an explanation in the associated legend.

Response: According to your suggestion, the explanation of the abbreviations “SE” and “B” has been added in the associated legend of table 3 (page 9).